# Grocery shopping as an outcome measure: A scoping review

Keith R. Cole[1,2,3]*, Sophie Minick[1,2], Leslie F. Davidson[4]

1 Department of Orthopaedic Surgery, Vanderbilt University Medical Center, Nashville Tennessee, United States of America, 2 Vanderbilt Center for Musculoskeletal Research, Vanderbilt University Medical Center, Nashville Tennessee, United States of America, 3 Department Health, Human Function, & Rehabilitation Sciences, George Washington University School of Medicine and Health Sciences, Washington District of Columbia, United States of America, 4 Department of Clinical Leadership & Research, George Washington University School of Medicine and Health Sciences, Washington District of Columbia, United States of America

* keith.cole@vumc.org

## Abstract

### Background

Grocery shopping is a complex Instrumental Activity of Daily Living (IADL) requiring cognitive and physical components that can be used to assess functional performance. Real-world physical and cognitive demands of grocery shopping occur simultaneously; however, many existing outcome measures only include a single domain or subtask. The objective of this review is to examine how grocery shopping as a whole, or multiple simultaneous subtasks of grocery shopping, has been used as a functional outcomes measure.

### Methods

Peer-reviewed manuscripts in the English language were retrieved from PubMed, Medline, Scopus, and Cochrane published on or before December 20, 2024. Articles were included if an outcome measure included multiple subtasks of grocery shopping and excluded if an outcome was related to only a single subtask of grocery shopping. Extracted data included Author(s), Publication Title, Publication Year, Study Location, Study population, Grocery Setting (Virtual, Real-World, Simulated, or Patient Reported Answers), Grocery-specific measure, and Grocery-Specific outcome results.

### Results

Fifty-eight studies were included from 15 different countries. The most common populations studied were healthy adults (15) and psychiatric disorders (15). Common methods of assessment included patient-reported outcome measures (22), virtual reality (17), and physically simulated or real grocery shopping (20). Only three

**Data availability statement:** Data is found in the Supporting information.

**Funding:** National Institutes of Health Grant Number: 5R21AG077404 (KC). The funder did not play a role in the study design, data collection and analysis, decision to publish, or preparation of the manuscript.

**Competing interests:** The authors have declared that no competing interests exist.

studies examined naturalistic, free-living grocery shopping. Outcomes were related to cognitive functioning (28), physical or motor impairments (23), or behavioral aspects of shopping (9).

## Conclusions

This review provides critical insights into how grocery shopping has been adopted as a performance outcome measure across populations and testing environments. Despite the growing recognition of grocery shopping as a useful measure, gaps remain in the literature, especially related to a lack of studies that integrate cognitive and physical domains or explore its use in populations with combined cognitive and physical impairments.

---

## Introduction

Grocery shopping is an important Instrumental Activity of Daily Living (IADL) that requires both complex cognitive and physical functioning. The American Occupational Therapy Association [1] differentiates IADLs from ADLs as being more complex and related to supporting independent activities in the home or community. Examples of other IADLs include financial management (e.g., balancing a checkbook), driving, or meal preparation. Grocery shopping involves complex cognitive demands related to planning food items to purchase, navigating the store to locate each of the items they wish to purchase, selecting the needed item amongst different choices (size, cost, etc.), budgeting for the purchases, and utilizing high-level attention skills necessary for focus in a dynamic environment. Physical demands for the IADL of grocery shopping are also complex, requiring individuals to physically navigate physical obstacles that may occur in the community and the store, as well as performing potentially physically demanding tasks of lifting and carrying grocery items or pushing a cart. The combined cognitive and physical complex demands associated with grocery shopping, in addition to the familiarity of the task for the majority of the population, render it a useful activity for healthcare providers to examine when assessing the effects of a wide range of health conditions on an individual's life skill performance.

The significance of the complex demands of grocery shopping is underscored by the numerous functional measures and outcomes that incorporate different grocery shopping subtasks. For example, physically demanding subtasks are incorporated in patient-reported outcome measures (PROMs) such as the 36-Item Short Form Survey (SF-36) [2] and the Patient-Reported Outcomes Measurement Information System (PROMIS) Global Health and Physical Function scales [3,4], which ask individuals about limitations in their ability to lift and/or carry groceries. Both the SF-36 and PROMIS measures have been validated across a wide range of populations, including those experiencing aging, schizophrenia, and neurodegenerative diseases [5–10]. Physical subtasks are also incorporated into functional tests such as the Grocery Shelving Task, where individuals are assessed for their ability to carry and place items on a shelf at shoulder height [9,10]. Additionally, cognitively demanding

subtasks are often incorporated into standardized testing. Verbal learning and memory abilities are assessed with measures such as the California Verbal Learning Test [11] and the International Shopping List Test [12]. These tests involve presenting grocery items to an individual and asking them to recall the items after a specified duration. These tests utilize the ecological relevance of establishing and remembering a shopping list. Verbal memory tests such as these have been shown to be sensitive in the ability to detect differences in cognitive function in conditions such as Alzheimer's disease [13,14] and mild traumatic brain injury [14].

Testing paradigms may also incorporate two simultaneous subtasks of grocery shopping. Cognitive-motor dual-task (CMDT) tests mimic real-world demands of moving and thinking by asking an individual to perform a movement task. One such CMDT involves the common grocery subtasks of walking (motor task) while recalling a shopping list (recall task) [15–17]. The difference in single vs. dual-task performance, termed the dual-task effect, exposes the impact that real-world demands of shopping may have on cognitive and motor functions associated with shopping. CMDT impairments have also been shown to be related to a number of health conditions, such as increased risk for falls [18] and mild cognitive impairment [19].

Measuring the ability to perform subtasks of grocery shopping has shown promise in detecting functions associated with health conditions, either individually (e.g., grocery shelving task) or as a multi-domain assessment (e.g., SF-36). However, it is the complex integration of multiple simultaneous cognitive and/or physical subtasks performance that results in accomplishing an IADL. This scoping review summarizes the literature that adopts grocery shopping assessments as a whole or those that include multiple subtask components. In other words, this review specifically considers studies that measure the totality of grocery shopping as opposed to isolating and measuring a single task or element of the shopping experience. Detection of declines in complex whole IADLs, such as grocery shopping, may be important for identifying individuals at risk of losing independence, as reflected in decreased performance in important daily living tasks. Populations who may benefit greatest might be those with prodromal dementia [20]. The National Institute on Aging (NIA) defines dementia as a loss of function significant enough to impair independence in daily activities, or a decline in neuro-motor function that may necessitate additional or increased physical or cognitive assistance. This may be especially relevant as technology that can track physical and cognitive functioning during free-living real-world activities improves. These advancements allow measures of the complex IADL of grocery shopping to be potentially useful as a digital biomarker of function.

This scoping review's primary objective is to describe the current literature that uses multiple subtasks of grocery shopping, or the entire activity, as an outcome measure. By describing the literature, we aim to identify methods and tools that measure grocery shopping ability and the populations in which they are measured. This scoping review will also identify any gaps in the literature or opportunities for future research that may improve the ability to use grocery shopping as a functional measure.

## Methods

This scoping review was conducted in accordance with the PRISMA-ScR framework, established by the Joanna Briggs Institute and further developed by Tricco et al [21]. We conducted a search for studies in peer-reviewed manuscripts with original results written in English from databases of PubMed, Medline, Scopus, and Cochrane on or before December 20, 2024. Searches combined the terms grocery or "food shopping" and function. An initial search revealed a significant number of studies that examined diet, nutrition, and access to food or grocery stores. These topics were out of the scope of the current review and a follow-up search was performed using the NOT Boolean term. An example search strategy for PubMed is: (grocery OR "food shopping") AND function NOT (diet OR access OR nutrition). Inclusion criteria required studies that 1) involve grocery shopping either physically, virtually, or patient-reported questions about shopping abilities, and 2) use grocery shopping as an assessment of function, including domains of cognition, motor performance, physical health, or behavioral health. Studies were excluded if they 1) used a grocery list of items as a portion of a non-grocery

shopping task (e.g., dual-task walking), 2) Assessed an individual's nutrition or diet, 3) assessed access and barriers to food and grocery stores, and 4) examine societal mobility.

Covidence systematic review software (Veritas Health Innovation, Melbourne, Australia) was used by two independent reviewers to first screen an exhaustive list of titles and abstracts of articles (n = 669). All conflicts were solved by discussion and consensus between the reviewers. Ultimately, 530 articles were excluded during the title and abstract review, where the remaining 139 were retrieved for further screening. Full-text articles were then screened in the same manner, although five articles were not able to be retrieved resulting in 134 full-text assessments. Following screening, 58 full-text articles met inclusion and exclusion criteria, as agreed upon by independent reviewers, and were included for data extraction. Fig 1 represents the PRISMA flow diagram for this review. Data was extracted and compiled in Microsoft Excel (Microsoft 365 MSO, Version 2406) by one author and then examined and verified by all other authors. Data variables included Author(s), Publication Title, Publication Year, Study Location, Study population, Grocery Setting (Virtual, Real-World, Simulated, or Patient Reported Answers), Grocery-specific measure, and Grocery-Specific outcome results. See S1 Table for full extraction.

## Results

### Country of origin

The search and screening process yielded 58 publications included for extraction and data analysis. Though this review was limited to literature published in the English language, the results spanned many different countries (Fig 2, top left panel). This included 35 (60%) manuscripts from the United States [22–53], six (10%) from Canada [54–59], three (5%) from Belgium [60–62], two (3%) each from China [63,64], England [65,66], France [67,68], and Germany [69,70], and one (2%) each from Australia [71], Colombia [72], Czech Republic [73], Israel [74], Sweden [75], and Switzerland [76].

### Study populations

There was a wide variability in the populations studied (Fig 2, top bottom panel). The most common populations investigated were healthy adults and those with psychiatric disorders, which included 15 (29%) studies each. Of the healthy adult studies, eight (53%) included only older adults [22,23,25–29,66], four (27%) included only young adults [30,31,69,73], two (13%) included younger and older adults [50,65], and one (7%) included only mothers of a child under five years old [71]. Studies involving those with psychiatric disorders included seven (47%) with diagnoses of multiple serious mental illnesses [39,40,44–47,74], five (33%) with schizophrenia or schizoaffective disorder [41–43,56,62], and one (7%) each with only major depression [48], obsessive compulsive disorder [70], or bipolar disorder [61].

Other populations of study included eight (14%) involving those with neurocognitive disorders [37,49,51,53,55,64,67,68], seven studies (12%) on musculoskeletal and mobility disorders [24,33–36,75,76], three (5%) on persons who had a stroke [57,58,77], two (3%) each involving people with neurodevelopmental disorders [38,63], general surgical patients [32,54], vestibular disorders [52,78] and vision deficits [72,79], and one study (2%) on those with traumatic brain injury [59], and substance dependence [60].

### Assessment delivery method

We found that assessments were delivered in multiple ways (Fig.2, top right panel). Twenty-two studies (38%) used a patient-reported outcome measure (PROM) to assess patients' perceptions of grocery shopping ability. Eight standardized questionnaires were used, where three used the IADL Profile [26,27,59], three used the Lawton IADL scale [28,54,72], two used the Multi-Level Assessment IADLS Scale [29,48], and one study each used the MILES Self-Report Questionnaire [67], NOMO 1.0 (Nordic Mobility Instrument) [75], and the Social Outcomes Questionnaire [77]. The remaining 11 studies used a custom IADL [22,23,66,70,79] or only asked a specific question related to grocery shopping ability or independence [25,32–35,76].

**Grocery Shopping**

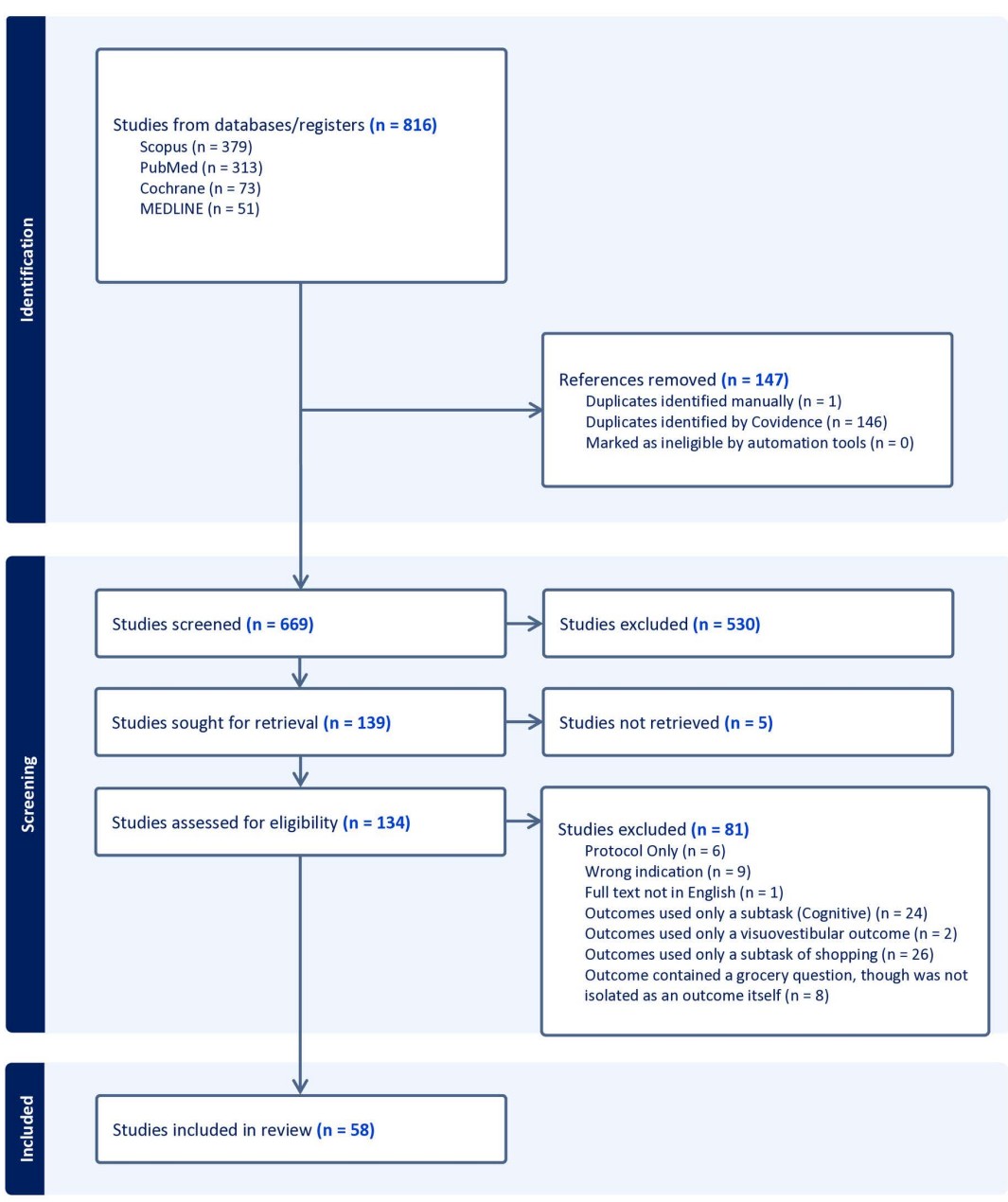

18th November 2024

covidence

**Fig 1. Flow chart for screening procedures (Covidence).**

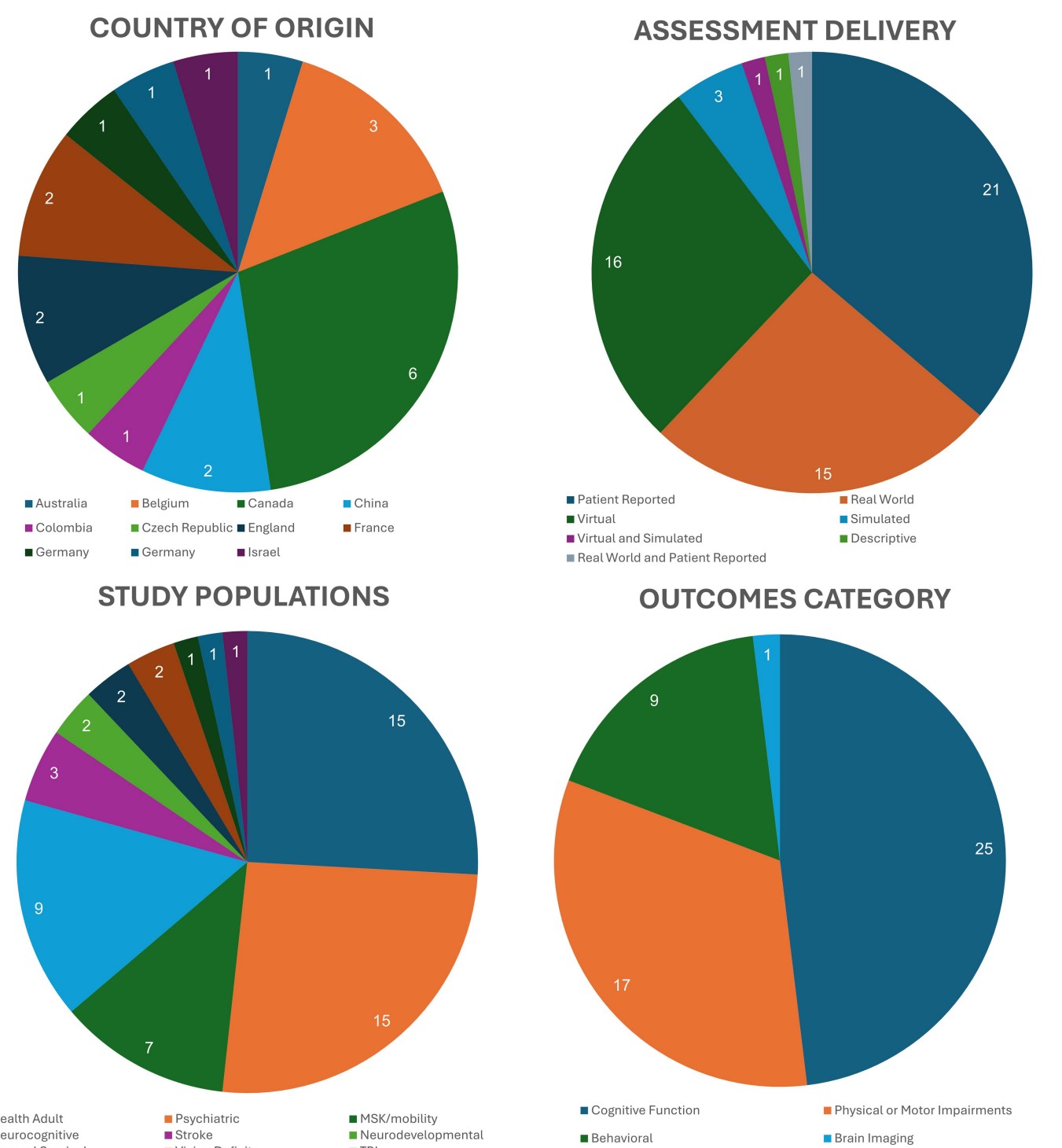

**Fig 2. Frequency of categories for each extracted data element.**

Twelve studies (21%) used virtual reality in a seated position to assess grocery shopping abilities [30,31,37,49,56,57,60–63,65,78], including using a controller to navigate through a virtual store and recognizing and selecting grocery items. One test [64] required the participant to provide a detailed verbal description of shopping-related details when shown pictures of a meal to prepare and the layout and pictures of the aisles of an unfamiliar grocery store.

Twenty studies (34%) assessed grocery shopping while physically moving through an environment. Of these studies, sixteen (80%) tested participants in a real-world grocery store [24,38–47,53,55,71,74], and four (20%) took place in a simulated store environment in a lab [36,53,68,69]. Instead of physically moving through a store, five (9% of all studies) used virtual reality environments [50–52,58,73]. Of the 16 studies that assessed shopping performance in a real grocery store, 12 (75%) included structured shopping tasks described and measured, of which 10 used the Test of Grocery Shopping Skills [39–47,74]. Only three (5% of all extracted studies) examined naturalistic, free-living grocery shopping activities [24,44,71].

## Study outcomes

Although the heterogeneous nature of the studies included in this scoping review did not allow for meta-analysis, we did investigate the themes of the major findings of each study (Fig 2, bottom right panel). The most common outcome of using grocery shopping as a measure was related to cognitive function and was the emphasis of 28 studies [26,30,31,37–39,41–43,46–51,53,55,57,59–62,64,65,67,68,76,77]. Twenty-three studies investigated grocery shopping related to physical impairments or motor function [22,23,25,27,29,32,34–36,40,49–52,54,56,58,69,71,73,75,78,79]. Behavioral aspects of grocery shopping were examined in nine different studies [24,28,33,44,45,66,70,72,74]. Only one article examined brain imaging related to grocery shopping ability [77].

## Discussion

This scoping review describes the body of evidence that uses grocery shopping as an outcome measure. Grocery shopping was used to measure function in a variety of health conditions, with the most common including aging, psychiatric disorders, musculoskeletal and mobility disorders, and neurocognitive disorders. Assessments were commonly used to describe or were related to cognitive or physical/motor function. Grocery shopping assessments were most commonly performed using PROMs, with grocery shopping isolated as a primary outcome, although a similar number of studies used virtual or physical grocery stores to assess shopping performance.

An interesting finding of this scoping review was that individuals with psychiatric disorders were a population where grocery shopping ability was studied frequently. Psychiatric diagnoses included serious mental illness, bipolar disorder, obsessive compulsive disorder, and schizophrenia or schizoaffective disorder. Of the studies including individuals with psychiatric disorders, nine emphasized the relationship between cognitive function and grocery shopping ability and how cognitive functioning in this population relates to being able to function independently in complex real-world tasks [39–43,46,47,61,62]. Ten studies in those with psychiatric disorders used the Test of Grocery Shopping Skills (TOGSS) [39–47,74], a validated measure of structured grocery shopping that occurred in a real-world grocery store. The TOGSS requires a trained occupational therapist to observe and score the accuracy and specific behaviors of the individual performing the test. Emphasis is placed on finding the correct shopping items as well as redundant behaviors associated with attempting to find each item. Interestingly, one study used eye tracking in those with serious mental illness to show that although task planning is similar to a control population, the number of fixations and scanning patterns were different, offering a possible explanation for declines in accuracy and efficiency [74]. Although cognitive functioning and overall independence are examined during this test, physical impairments or motor control contributions to grocery shopping are not considered or assessed.

Healthy older and younger adults were also frequently assessed. In studies involving healthy older adults, the most frequent method of assessing grocery shopping was patient-reported outcome measures [22,23,25–29]. As required in the

inclusion criteria, PROM questions asked a person's ability to perform grocery shopping rather than a subtask of grocery shopping (e.g., carrying or lifting groceries). Although most PROMs asked questions related to several different IADL and/or ADLs, outcomes of each of the reported studies were at least related to the specific item asking about their ability to perform grocery shopping. Of these studies, only one found that those with low, but within the range of normal, cognitive function had a greater likelihood of having limitations in grocery shopping [65]. Five studies, however, examined the relationship between physical function/limitations and the ability to grocery shop. These studies described relationships between upper extremity strength [22,29], vision impairment [27], risk of falls [25], and risk of limitation in specific ADLS (ADLs) related to independence in grocery shopping [23]. Three other studies involving older adults used performance-based virtual grocery shopping tasks. These tasks demonstrated significant declines in performance between older and younger adults [50], and demonstrated that virtual performance was associated with better adaptive functioning in older adults [49].

Interestingly, five out of six studies that included healthy younger adults utilized virtual environments for assessments of grocery shopping [30,31,50,65,73]. These studies indicated that in younger adults, there is a negative association between cognitive functioning and shopping errors and time, and gender may influence shopping performance. The two studies comparing older and younger adults assessed grocery shopping with virtual environments found that during shopping, older adults walk slower, have more errors, and are more vulnerable to distractions than younger adults [50,65].

Studies that involved populations with neurocognitive disorders [37,49,51,53,55,64,67,68] included those with an elevated risk of Alzheimer's disease, mild cognitive impairment, dementia (frontal, Alzheimer's, or undefined), and Parkinson's disease (PD). Each of the studies investigated the relationship between cognition and grocery shopping, though one study examined the influence of motor changes in those with PD [51]. The methods for measuring grocery shopping ability for this population, however, varied considerably. One study each used an observer-rater structured shopping episode [55], a structured shopping task in a real grocery store compared to a PROM [67], a verbal description of all of the steps that are involved in performing a grocery shopping task [64], a combined verbal description followed by execution of tasks to complete grocery shopping in a real store [68], and a virtual reality shopping task [37]. These studies provide insights into a wide range of functions related to cognition, indicating that although there is significant heterogeneity in awareness of perceived vs. real grocery shopping, executive functioning and verbal learning are more critical for grocery shopping performance than semantic memory. It is interesting to note that physical function and performance were considered in only one study despite the breadth of literature that describes altered gait, balance, and increased risk of falls in older adults with mild to severe cognitive impairments [80–83]. This study demonstrated that, even though no difference was found for traditional motor and cognitive measures compared to controls, those with PD had slower gait speed while reading a shopping list and spent more time turning and stopping [51].

Participants that had musculoskeletal disorders and mobility impairments included five studies that used a patient-reported question specific to grocery shopping [33–35,75,76]. These studies provide evidence that mobility devices (e.g., scooters) increase a person's independence in grocery shopping. These studies also report findings that reported grocery shopping ability at baseline and two months following total hip arthroplasty is related to pain and one-year functional outcomes. Only one study in this population considered cognitive effects [76], demonstrating that asking about grocery shopping ability strongly correlated with cognitive status and was helpful to surgeons in deciding appropriate care for individuals seeking treatment for femoral neck fractures. One study video recorded real-world grocery shopping episodes of those with mobility impairments [24]. This study found that those with higher levels of mobility impairment demonstrated significantly different physical behaviors specific to carrying less weight, turning their head fewer times, and performing fewer postural transitions (reaching, bending, etc.). Although specific measures of dynamic balance were not measured, this study emphasizes the different movement strategies that those with mobility impairments might use to perform the complex IADL of grocery shopping.

### Gaps in the literature

Despite the fact that grocery store shopping has been used in a wide variety of patient populations as an outcome measure, there are noticeable gaps in the literature. The small number of studies (3 or fewer) in several populations

(e.g., stroke, traumatic brain injury) and their heterogeneous methodology rendered comparison of studies within these populations difficult. Although we found PROMs that included grocery shopping as a question item and found many unique structured, virtual, and real-life grocery tasks, we found only one validated assessment of grocery shopping as an outcome measure for those with psychiatric disorders (TOGSS). Using ecologically valid IADL assessments incorporating familiar tasks such as grocery shopping may provide much-needed information regarding performance changes as they relate to the complex demands people experience in everyday life activities. While grocery shopping involves simultaneously performing complex physical and cognitive subtasks, there is a paucity of studies that consider how both physical and cognitive functioning relate to grocery shopping performance. Measures such as PROMs cannot disentangle the relationship between motor and cognitive functioning. Simulated or virtual activity engagement eliminates multiple real-world variables that may potentially shape grocery shopping ability. Finally, when structured or free-living grocery shopping tasks are utilized, the majority of studies examine the cognitive (item selection) and navigation aspects (wayfinding) of shopping. There is a need to measure motor performance with greater precision, such as posture, balance, stride, reaching performance, and cadence, to identify physical impairments that might contribute to shopping abilities.

A grocery store is potentially an ideal environment to detect signatures of functional decline in both motor and cognitive domains. Wearable technology has not been utilized for grocery shopping assessments, but has the potential to measure precise digital biomarkers related to combined motor and cognitive function outcomes. Finally, and most markedly, there is a lack of ecologically valid functional assessments for people on the spectrum of Alzheimer's disease and related dementias, even though the definition of dementia includes the loss of ability to perform ADLs/IADLs.

## Strengths and limitations

There are several strengths of this scoping review. The review followed PRISMA-ScR guidelines for scoping reviews [21]. This scoping review also included a broad range of assessment methods for performing grocery shopping among diverse populations. A strength of this was including only studies that measured more than just one or two sub-tasks of grocery shopping, therefore assessing the imbedded complexity of cognitive and physical functioning to perform the important IADL.

There are also several limitations to this study. The studies reviewed varied significantly in methodology, population, and assessment tools, which limits the ability to make direct comparisons between studies and renders meta-analysis or quantitative synthesis impractical. The review was only performed in the English language, which may have led to an incomplete synthesis of data if studies were performed in non-native English-speaking countries and limited the generalizability of our findings to cultures that are not primarily English-speaking. We also did not specifically search for gray literature and may have missed work published that is not cross-referenced in the databases in the search strategy, both of which may limit the scope of articles included in extraction.

## Conclusions

This scoping review highlights the role that grocery shopping and multiple simultaneous grocery shopping subtasks can play as a measure in assessing cognitive and physical capabilities across a range of health conditions and ages. The complex and integrative nature of grocery shopping, involving both cognitive and physical skills, makes it a valuable activity for understanding the functional status of individuals. The literature reviewed demonstrates a variety of measures that adopt grocery shopping from patient-reported outcomes to virtual and real-world tasks.

The findings suggest that while grocery shopping assessments are commonly used to measure cognitive and physical functions, there is a noticeable gap in studies that combine both aspects. As technology advances, there is an opportunity to enhance the assessment of grocery shopping through wearable devices and other tools that can monitor cognitive and physical performance in real-time. Such developments could lead to the use of grocery shopping as a digital biomarker for functional decline, particularly in aging populations or those with progressive neurological conditions.

Overall, this scoping review underscores the potential of grocery shopping as a multifaceted outcome measure that can provide critical insights into an individual's ability to perform daily living activities independently. Future research should focus on developing more comprehensive assessment tools that integrate cognitive and physical domains, thereby improving the detection of functional decline and informing interventions to maintain independence in at-risk populations.

## Supporting information

**S1 Table. Data extracted from all manuscripts that met inclusion and exclusion criteria.**
(XLSX)

## Author contributions

**Conceptualization:** Keith Cole, Leslie Davidson.

**Data curation:** Keith Cole, Sophie Minick, Leslie Davidson.

**Formal analysis:** Keith Cole.

**Funding acquisition:** Keith Cole.

**Investigation:** Keith Cole, Sophie Minick, Leslie Davidson.

**Methodology:** Keith Cole, Leslie Davidson.

**Project administration:** Keith Cole, Leslie Davidson.

**Resources:** Keith Cole.

**Software:** Keith Cole.

**Supervision:** Keith Cole.

**Writing – original draft:** Keith Cole.

**Writing – review & editing:** Keith Cole, Sophie Minick, Leslie Davidson.

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
