## [Decision Letter · Decision Letter 0]

11 Apr 2025

PONE-D-25-03026Grocery shopping as an outcome measure: a scoping reviewPLOS ONE

Dear Dr. Cole,

Thank you for submitting your manuscript to PLOS ONE. After careful consideration, we feel that it has merit but does not fully meet PLOS ONE’s publication criteria as it currently stands. Therefore, we invite you to submit a revised version of the manuscript that addresses the points raised during the review process.

We look forward to receiving your revised manuscript.

Kind regards,

Rae Yule Kim

Academic Editor

PLOS ONE

**Journal Requirements:**

1. When submitting your revision, we need you to address these additional requirements. Please ensure that your manuscript meets PLOS ONE's style requirements, including those for file naming. The PLOS ONE style templates can be found at https://journals.plos.org/plosone/s/file?id=wjVg/PLOSOne_formatting_sample_main_body.pdf and https://journals.plos.org/plosone/s/file?id=ba62/PLOSOne_formatting_sample_title_authors_affiliations.pdf

**Additional Editor Comments:**

Please find the attached reviews. The majority of reviewers are favorable and agree that the paper makes a meaningful contribution, warranting publication in PLOS ONE pending minor revisions, most of which can be addressed through copyediting. In my assessment, the manuscript offers a valuable discussion on the potential of grocery shopping behavior as an indicator of cognitive impairment and related health conditions.

Please address each of the reviewers' comments and submit the revised manuscript within one month. With your revision, please attach the original, high-resolution file for Figure 2 to ensure improved image quality in the final publication.

Reviewers' comments:

Reviewer's Responses to Questions

**Comments to the Author**

1. Is the manuscript technically sound, and do the data support the conclusions?

Reviewer #1: Partly

Reviewer #2: Yes

Reviewer #3: Yes

2. Has the statistical analysis been performed appropriately and rigorously? 

Reviewer #1: No

Reviewer #2: Yes

Reviewer #3: N/A

3. Have the authors made all data underlying the findings in their manuscript fully available?

Reviewer #1: Yes

Reviewer #2: Yes

Reviewer #3: Yes

4. Is the manuscript presented in an intelligible fashion and written in standard English?

Reviewer #1: Yes

Reviewer #2: Yes

Reviewer #3: Yes

5. Review Comments to the Author

**Reviewer #1: ** Thank you for the opportunity to review your manuscript titled :Grocery Shopping as an Outcome Measure: A Scoping Review.

Your study presents a well-structured and comprehensive review of how grocery shopping has been used as a functional outcome measure across different populations and testing environments. I provide my detailed feedback on the manuscript, addressing its strengths and areas that require improvement.

The study follows a scoping review methodology, ensuring a transparent and replicable research process.The review includes 58 studies from 15 different countries, covering a wide range of populations.The study highlights the cognitive and physical complexity of grocery shopping, reinforcing its potential as a functional assessment tool.The conclusions are largely supported by the reviewed literature, particularly regarding the relationship between cognitive decline, psychiatric disorders, and functional independence.

Areas for Improvement:

1.The studies reviewed vary significantly in methodology, population, and assessment tools, making direct comparisons difficult. While this is a limitation of scoping reviews, it should be explicitly acknowledged.

2. The study does not conduct a meta-analysis or any quantitative synthesis, which limits the ability to draw strong numerical conclusions about trends across studies. Including a summary of key findings in percentages would enhance readability.

3.The Data Availability Statement confirms that all relevant data are included in the manuscript and supporting files.

Supplemental Table 1 provides extracted study data, ensuring transparency. If possible, include a table summarizing key study findings numerically to improve data interpretation.

**Reviewer #2: ** This great scoping review describes the current literature that uses multiple subtasks of grocery shopping as an outcome measure. The authors adequately searched the major databases and selected 58 studies to characterize the study population, describe assessment delivery methods, and describe study outcomes.

Major issues

In the results, I couldn´t find references to the studies of Ogourtsova et al., 2018; Kizony et al., 2010 and Chang et al., 2022. in the study population section (lines 158-172). In the assessment delivery method section, I couldn´t find references to the studies of McGuire et al., 2006 and Bosley et al., 2004.

Since these papers meet the inclusion criteria and were described in the country of origin section I think they should be described and cited in all other results sections.

Minor issues

In the introduction, the sentence "Both the SF-36 and PROMIS measures have been validated across a wide range of populations, including those experiencing aging, schizophrenia, and neurodegenerative diseases.[5, 6]" (lines 68-70) refers to the '36-Item Short Form Health Survey questionnaire' as SF-36, I think it is best to use the full name in this first reference and maintain the acronym in the next ones. In this same sentence, the author mentions populations experiencing aging, schizophrenia, and neurodegenerative diseases, however, the work of Hill et al., 2001 and Shields et al., 2008 (references 5 and 6) describes patients with chronic obstructive pulmonary disease and adults with Down syndrome, respectively. I think it would be better if the authors could add references to populations experiencing aging, schizophrenia, and neurodegenerative diseases. Also, in line 70, and several other lines, the reference number is placed after the comma, the authors should carefully scan the paper to uniform this.

In the introduction, the sentence "Verbal memory tests such as these have 79 been shown to be sensitive in the ability to detect differences in cognitive function in 80 conditions such as Alzheimer’s disease[9] and mild traumatic brain injury [9]." (lines 78-80) mention populations experiencing Alzheimer’s disease and mild traumatic brain injury, but the work of Leitner et al., 2019 describes a heterogeneous sample of people with traumatic brain injury. It would be better if the authors could add a reference directly related to populations experiencing Alzheimer’s too.

The introduction section has excellent content, but, if the authors can group the related concepts and cut some words, it will be even better.

In the discussion, the sentence "Of these 15 studies, nine emphasized the relationship between cognitive function and grocery shopping ability and how cognitive functioning in this population relates to being able to function independently in complex real-world tasks.[34-38, 41, 42, 56, 57]" (lines 221-224), I understand that by "these 15 studies" the authors mean the studies with individuals with psychiatric disorders. However, since the number 15 was used to refer to these studies only in the results section, maybe it's best to repeat something like "studies with individuals with psychiatric disorders" to make it clear. Don't forget to place the reference number before the comma.

I found the sentence "These studies provide evidence that mobility devices (e.g., scooters) increase a person’s independence in grocery shopping and reported grocery shopping ability at baseline and two months following total hip arthroplasty is related to pain and one-year functional outcomes." (lines 284-287) a little bit hard to understand. I think it will be better If it can be divided into two small sentences.

The sentence "Finally, and most markedly, there is a lack of ecologically valid functional assessments for people on the spectrum of ADRD, even though the definition of dementia includes the loss of ability to perform ADLs/IADLs." (lines 322-324) Maybe the authors should use 'Alzheimer's disease and related dementias' before using ADRD to make the paper more readable to people who are not from the field.

**Reviewer #3:**  In Figure 2, in the study population, instead of adult, write healthy adult.

In the discussion, line 213, I think the correct word is neurocognitive instead of neurodevelopmental.

In the study population, line 165 mentions a study that uses only a population with schizophrenia; but, in the discussion, line 221 mentions schizophrenia or schizoaffective disorder, it is not clear since they are two different disorders.

In figure 2, the study population includes a study with people who have suffered a stroke, however, this is not mentioned in the results or in the discussion.

For a better understanding of the graphs in figure 2, it would be good to include the number of studies within the graph.

6. PLOS authors have the option to publish the peer review history of their article (what does this mean? ). If published, this will include your full peer review and any attached files.

**Do you want your identity to be public for this peer review?** For information about this choice, including consent withdrawal, please see our Privacy Policy .

Reviewer #1: No

Reviewer #2: **Yes: ** Daniella F P C A Durço

Reviewer #3: **Yes: ** MARIA JOSE IRIAS ESCHER

---

## [Author Response · Author response to Decision Letter 0]

16 Apr 2025

Response to Reviewers

We thank the reviewers and editor for your thoughtful appraisal and comments.

Each reviewer-suggested change is listed and immediately followed by our response.

Reviewer #1:

1. The studies reviewed vary significantly in methodology, population, and assessment tools, making direct comparisons difficult. While this is a limitation of scoping reviews, it should be explicitly acknowledged. The study does not conduct a meta-analysis or any quantitative synthesis, which limits the ability to draw strong numerical conclusions about trends across studies.

We completely agree with your statement. We stated in the Study outcomes section previously “Although the heterogeneous nature of the studies included in this scoping review did not allow for meta-analysis, we did investigate the themes of the major findings of each study” We acknowledge that this is not a complete assessment of the limitations. We have now added the following sentence to the limitations section. “The studies reviewed varied significantly in methodology, population, and assessment tools, which limits the ability to make direct comparisons between studies and renders meta-analysis or quantitative synthesis impractical.“

2. Including a summary of key findings in percentages would enhance readability.

Thank you for this suggestion. We have now included percentages through the methods section along with the study counts for each category discussed.

3. Supplemental Table 1 provides extracted study data, ensuring transparency. If possible, include a table summarizing key study findings numerically to improve data interpretation.

Thank you for this suggestion. Also in alignment with a comment from Reviewer 3, we have addressed this by including the number of studies for each result category in the pie chart, which hopefully will improve the ability for readers to quantify the subgroups that we were able to discuss in the results and discussion sections of this paper.

Reviewer #2:

1. In the results, I couldn´t find references to the studies of Ogourtsova et al., 2018; Kizony et al., 2010 and Chang et al., 2022. in the study population section (lines 158-172). In the assessment delivery method section, I couldn´t find references to the studies of McGuire et al., 2006 and Bosley et al., 2004. Since these papers meet the inclusion criteria and were described in the country of origin section I think they should be described and cited in all other results sections.

Thank you for catching this oversight! We absolutely agree that these studies should also be included in these sections. They have now been included in each section. In explanation, we inadvertently left stroke and the 3 associated citations out of the study population section. Citation errors were made in the assessment delivery, which have now been corrected. 1) McGuire 2006 is referenced for “IADL Profile” instead of Bier et al., 2013. In the same section, Bier et al., was accurately referenced for testing in a real-world grocery store in the previous draft. 2) Bosley et al., 2004 is now referenced in the statement of “only asked a specific question related to grocery shopping ability or independence” instead of Kiosses et al., 2005. Kiosses et al., 2005 is referenced correctly for the Multi-Level Assessment IADLS scale.

2. In the introduction, the sentence "Both the SF-36 and PROMIS measures have been validated across a wide range of populations, including those experiencing aging, schizophrenia, and neurodegenerative diseases.[5, 6]" (lines 68-70) refers to the '36-Item Short Form Health Survey questionnaire' as SF-36, I think it is best to use the full name in this first reference and maintain the acronym in the next ones.

Thank you for bringing this to our attention. This has now been corrected in the previous sentence where they first appear.

3. In this same sentence, the author mentions populations experiencing aging, schizophrenia, and neurodegenerative diseases, however, the work of Hill et al., 2001 and Shields et al., 2008 (references 5 and 6) describes patients with chronic obstructive pulmonary disease and adults with Down syndrome, respectively. I think it would be better if the authors could add references to populations experiencing aging, schizophrenia, and neurodegenerative diseases.

Thank you for bringing this oversight to our attention. This is now updated with references for these specific populations.

4. Also, in line 70, and several other lines, the reference number is placed after the comma, the authors should carefully scan the paper to uniform this.

Thank you for catching this inconsistency. Reference numbers are now placed before all commas and periods.

5. In the introduction, the sentence "Verbal memory tests such as these have 79 been shown to be sensitive in the ability to detect differences in cognitive function in 80 conditions such as Alzheimer’s disease[9] and mild traumatic brain injury [9]." (lines 78-80) mention populations experiencing Alzheimer’s disease and mild traumatic brain injury, but the work of Leitner et al., 2019 describes a heterogeneous sample of people with traumatic brain injury. It would be better if the authors could add a reference directly related to populations experiencing Alzheimer’s too.

Thank you for catching this referencing error. This is now corrected with an appropriate reference to a study involving those affected by Alzheimer’s disease.

5. The introduction section has excellent content, but, if the authors can group the related concepts and cut some words, it will be even better.

We appreciate the comment and have reviewed the introduction in the context of the rest of the manuscript. We believe that the format of the current manuscript is important to demonstrate the multi-layered paradigm that is key to understanding dual and multi-tasking in an ecologically valid environment. If we group concepts together, the information does not reflect the complex nature of what we have been trying to understand. We hope that helps clarify.

6. In the discussion, the sentence "Of these 15 studies, nine emphasized the relationship between cognitive function and grocery shopping ability and how cognitive functioning in this population relates to being able to function independently in complex real-world tasks.[34-38, 41, 42, 56, 57]" (lines 221-224), I understand that by "these 15 studies" the authors mean the studies with individuals with psychiatric disorders. However, since the number 15 was used to refer to these studies only in the results section, maybe it's best to repeat something like "studies with individuals with psychiatric disorders" to make it clear.

Thank you for this suggestion. The statement has been changed to “Of the studies including individuals with psychiatric disorders, “

7. I found the sentence "These studies provide evidence that mobility devices (e.g., scooters) increase a person’s independence in grocery shopping and reported grocery shopping ability at baseline and two months following total hip arthroplasty is related to pain and one-year functional outcomes." (lines 284-287) a little bit hard to understand. I think it will be better If it can be divided into two small sentences.

Thank you for this suggestion. The sentence has been broken up into two sentences: “These studies provide evidence that mobility devices (e.g., scooters) increase a person’s independence in grocery shopping. These studies also report findings that reported grocery shopping ability at baseline and two months following total hip arthroplasty is related to pain and one-year functional outcomes.“

8. The sentence "Finally, and most markedly, there is a lack of ecologically valid functional assessments for people on the spectrum of ADRD, even though the definition of dementia includes the loss of ability to perform ADLs/IADLs." (lines 322-324) Maybe the authors should use 'Alzheimer's disease and related dementias' before using ADRD to make the paper more readable to people who are not from the field.

Thank you for the suggestion. This has been changed accordingly.

Reviewer #3:

1. In Figure 2, in the study population, instead of adult, write healthy adult.

Thank you for bringing this to our attention. This is now corrected.

2. In the discussion, line 213, I think the correct word is neurocognitive instead of neurodevelopmental.

Thank you for catching this typographical error. This is now corrected.

3. In the study population, line 165 mentions a study that uses only a population with schizophrenia; but, in the discussion, line 221 mentions schizophrenia or schizoaffective disorder, it is not clear since they are two different disorders.

Thank you for bringing this to our attention. Some of the included studies examined both schizophrenia and schizoaffective disorder in the same study. This statement has been revised to “schizophrenia or schizoaffective disorder[]”.

4. In figure 2, the study population includes a study with people who have suffered a stroke, however, this is not mentioned in the results or in the discussion.

We apologize; this is now included in the results in the “Study populations” section.

5. For a better understanding of the graphs in figure 2, it would be good to include the number of studies within the graph.

Thank you for this suggestion. We have now added the number of studies in each wedge of each pie chart.

---

## [Decision Letter · Decision Letter 1]

30 Apr 2025

Grocery shopping as an outcome measure: a scoping review

PONE-D-25-03026R1

Dear Dr. Cole,

We’re pleased to inform you that your manuscript has been judged scientifically suitable for publication and will be formally accepted for publication once it meets all outstanding technical requirements.

Kind regards,

Rae Yule Kim

Academic Editor

PLOS ONE

Reviewers' comments:

Reviewer's Responses to Questions

**Comments to the Author**

1. If the authors have adequately addressed your comments raised in a previous round of review and you feel that this manuscript is now acceptable for publication, you may indicate that here to bypass the “Comments to the Author” section, enter your conflict of interest statement in the “Confidential to Editor” section, and submit your "Accept" recommendation.

Reviewer #1: All comments have been addressed

Reviewer #2: All comments have been addressed

2. Is the manuscript technically sound, and do the data support the conclusions?

Reviewer #1: Yes

Reviewer #2: (No Response)

3. Has the statistical analysis been performed appropriately and rigorously? 

Reviewer #1: Yes

Reviewer #2: (No Response)

4. Have the authors made all data underlying the findings in their manuscript fully available?

Reviewer #1: Yes

Reviewer #2: (No Response)

5. Is the manuscript presented in an intelligible fashion and written in standard English?

Reviewer #1: Yes

Reviewer #2: (No Response)

6. Review Comments to the Author

Reviewer #1: The authors present a well-organized and insightful scoping review examining the use of grocery shopping as a complex instrumental activity of daily living in functional outcome assessments across diverse populations. The manuscript is well-structured and comprehensive, and the authors have clearly responded to prior reviewer suggestions with thoughtful revisions. Several aspects of the study are commendable, including the systematic application of the PRISMA-ScR methodology, inclusion of diverse study populations, and the emphasis on cognitive and physical multitasking demands inherent to grocery shopping.

Reviewer #2: (No Response)

7. PLOS authors have the option to publish the peer review history of their article (what does this mean? ). If published, this will include your full peer review and any attached files.

**Do you want your identity to be public for this peer review?** For information about this choice, including consent withdrawal, please see our Privacy Policy .

Reviewer #1: **Yes: ** CHIU PO EN

Reviewer #2: No

---

## [Editor Report · Acceptance letter]

PONE-D-25-03026R1

PLOS ONE

Dear Dr. Cole,

I'm pleased to inform you that your manuscript has been deemed suitable for publication in PLOS ONE. Congratulations! Your manuscript is now being handed over to our production team.

Kind regards,

on behalf of

Dr. Rae Yule Kim

Academic Editor

PLOS ONE